# Disrupting *Plasmodium* UIS3–host LC3 interaction with a small molecule causes parasite elimination from host cells

Sonali Setua [1], Francisco J. Enguita [1], Ângelo Ferreira Chora [1], Harish Ranga-prasad[2], Aparajita Lahree[1], Sofia Marques[1], Varadharajan Sundaramurthy [2] & Maria M. Mota [1]✉

The malaria parasite *Plasmodium* obligatorily infects and replicates inside hepatocytes surrounded by a parasitophorous vacuole membrane (PVM), which is decorated by the host-cell derived autophagy protein LC3. We have previously shown that the parasite-derived, PVM-resident protein UIS3 sequesters LC3 to avoid parasite elimination by autophagy from hepatocytes. Here we show that a small molecule capable of disrupting this interaction triggers parasite elimination in a host cell autophagy-dependent manner. Molecular docking analysis of more than 20 million compounds combined with a phenotypic screen identified one molecule, C4 (4-{[4-(4-{5-[3-(trifluoromethyl) phenyl]-1,2,4-oxadiazol-3-yl}benzyl) piperazino]carbonyl}benzonitrile), capable of impairing infection. Using biophysical assays, we established that this impairment is due to the ability of C4 to disrupt UIS3–LC3 interaction, thus inhibiting the parasite's ability to evade the host autophagy response. C4 impacts infection in autophagy-sufficient cells without harming the normal autophagy pathway of the host cell. This study, by revealing the disruption of a critical host–parasite interaction without affecting the host's normal function, uncovers an efficient anti-malarial strategy to prevent this deadly disease.

---

[1] Instituto de Medicina Molecular João Lobo Antunes, Faculdade de Medicina Universidade de Lisboa, 1649-028 Lisbon, Portugal. [2] National Centre for Biological Sciences, Tata Institute of Fundamental Research, Bengaluru, Karnataka, India. ✉email: mmota@medicina.ulisboa.pt

Autophagy is a conserved cell-autonomous catabolic process responsible for the degradation and recycling of intracellular components to maintain cellular homoeostasis. Upon initiation of canonical autophagy, a double membrane autophagosome forms around the cellular materials that need to be degraded. The sequential activation of ubiquitin-like protein-machinery such as Atg7 (E1-like), Atg3 (E2-like) and Atg12-Atg5-Atg16L1 (E3-like) complex associates LC3 (Microtubule-associated protein 1 A/1B-light chain 3; a member of Atg8 protein family) with the autophagosome isolation membrane. Membrane-bound LC3 binds autophagy adaptor proteins on the cargo which leads to engulfment of the cargo by the autophagosomal membrane followed by its lysosomal degradation[1]. Although initially demonstrated as a response to cellular stress, autophagy is now a well-established host defence mechanism that can substantially hinder the virulence of the intracellular pathogens[2]. However, intracellular organisms have evolved various sophisticated strategies to manipulate this host cell pathway to their advantage or escape their recognition and capture by the autophagy machinery, thus preventing pathogen elimination from the host cell[3–5].

Apicomplexan parasites such as *Plasmodium spp.*, *Toxoplasma*, and *Theileria* are a class of intracellular pathogens that are responsible for several important human diseases. As obligatory intracellular organisms, they maintain a complex interplay of host-pathogen interaction with autophagy playing a critical role in this process[5,6]. Among these, *Plasmodium spp.*, the causative agent of malaria, demands special attention due to the enormous global health and economic burden caused by this disease[7]. While precise figures for malaria's impact are not known, WHO estimates ~250 million cases and ~ 500 thousand deaths per year due to this disease[8].

Five *Plasmodium* species – *P. falciparum, P. vivax, P. ovale, P. malariae* and *P. knowlesi* – are known to cause human malaria. The first two species are responsible for most infections worldwide; *P. falciparum* results in more than 90% of the deaths; and *P. vivax* and *P. ovale* have dormant liver-stage parasites forms, called "hypnozoites", which can reactivate, or relapse, and cause malaria several months or years after the initial infection. *Plasmodium* poses a complex life cycle that involves a mosquito vector and the mammalian host as well as infection of various cell types within the same mammalian host. Liver-stage infection is the first and obligatory step of malaria infection in mammals where *Plasmodium* sporozoite matures into thousands of pathogenic merozoites within the confinement of the parasitophorous vacuole membrane (PVM)[9] inside hepatocytes. The PVM forms the critical host-parasite interface during liver-stage infection. Although traditionally termed as the silent stage of malaria infection, recent pieces of evidence proved that the host cell could sense *Plasmodium* parasites and trigger a response against them[10,11]. Indeed, liver-stage *Plasmodium* PVM gets decorated with LC3-containing autophagy-derived vesicles as well as LC3 binding proteins such as p62, NBR1, and NDP52 along with ubiquitin[12,13]. Moreover, vesicles positive for LAMP1, a lysosomal marker protein, proposed to be used as a nutrient source by the parasite for its fast development inside hepatocytes, surround *Plasmodium* throughout infection without fusing with the PVM[14]. Thus, parasites can evade the host autophagy response. While it has been proposed that parasite redirect PVM-associated LC3 towards the tubulovesicular network (TVN) in later stages of intrahepatic development to avoid elimination by the host cells[15], we have shown that *Plasmodium*'s PVM-resident protein UIS3 (Upregulated in infective sporozoites 3) interacts with LC3 from very early stages of infection and thereby prevents parasite elimination from the cells by the host autophagy machinery[11].

Here we identify a small molecule inhibitor of the UIS3-LC3 interaction that allows the host cell to eliminate the parasite during liver-stage infection. We show that the inhibitor directly interacts with the UIS3-LC3 complex. In doing so, it inhibits the parasite's ability to evade the host autophagy response, a crucial step for *Plasmodium* survival and subsequent multiplication in the hepatocytes. Such inhibitors of *Plasmodium* evasion mechanism from the host autophagy defence, in combination with existing antimalarials, hold great potential as an efficient therapeutic strategy for malaria.

## Results

**Identification of the small molecule C4 hampering host cell infection by *Plasmodium* sporozoites.** Having found that *P. berghei* and *P. falciparum* UIS3 binds to mouse and human LC3 and protecting the parasite from elimination by host cell autophagy[11], we hypothesise that disruption of this interaction by a small molecule would lead to parasite elimination and impaired infection. To start testing this hypothesis we performed virtual compound library screen (VLS) by in silico molecular docking analysis using the published X-ray structures of *P. falciparum* UIS3 and human LC3 together with more than 20 million compounds from an open-access library, ZINC (Fig. 1a). This analysis led to a selection of 21 compounds that have the potential to bind at the UIS3-LC3 interacting region. Among these compounds, 15 compounds were available for use (Supplementary Table 1). All selected compounds were next tested in a phenotypic screen (PHS, Fig. 1b). For the PHS, we used, in addition to the rodent wild-type (WT) *P. berghei* ANKA parasite strain (259cl2), a newly generated parasite line where the endogenous *P. berghei uis3* gene was replaced by the *uis3* gene from *P. falciparum* (marked with 2 HA tag), PfUIS3@Pb. Briefly, the gene encoding *P. falciparum* UIS3-HA was inserted in the *P. berghei* parasite (507cl1) into the *uis3* locus under the control of native 5′-and 3′ regulatory sequences (Supplementary Fig. 1a). Genotyping of PfUIS3@Pb confirmed the correct integration of the PfUIS3-HA expression cassette (Supplementary Fig. 1b). Immunofluorescence microscopy analysis of infected Huh7 cells showed co-localisation of PfUIS3-HA with another known PVM marker protein, UIS4, indicating the correct localisation of the PfUIS3-HA at the PVM (Supplementary Fig. 1c). Both WT and PfUIS3@Pb parasite lines behave similarly throughout the life cycle, including the number of salivary gland sporozoites (average of 30,000 sporozoites per mosquito), liver-stage infectivity (Supplementary Fig. 1d–f) as well as blood parasitaemia (Supplementary Fig. 1g) and disease progression (Supplementary Fig. 1h). The PHS was then performed using Huh7 cells infected with sporozoites from either parasite lines with or without adding each of the compounds at 10 μM concentration two hours (h) before sporozoite addition. Infection level was then determined by measuring the number of GFP-positive infected cells by flow cytometry, representing *Plasmodium* exo-erythrocytic form, EEF, (as both parasite lines express GFP). Toxicity of the compounds towards the host cell was determined by measuring host cell confluency using flow cytometry (Fig. 1c–e). In both screens for the two independent parasite lines, the compound C4 was identified as the best hit, responsible for the highest decrease in infection (Fig. 1c–e). Docking experiments predicted a binding pocket for C4 in the surface of UIS3 protein that partially overlaps the proposed UIS3-LC3 interacting region of both *P. falciparum* and *P. berghei* UIS3 (Fig. 1f, g; Supplementary Figs. 4 and 5). C4 belongs to the class of phenyloxadiazoles, which are polycyclic aromatic compounds containing a benzene ring linked to a 1,2,4-oxadiazole ring through a C-C or C-N bond. The oxadiazole ring is connected to

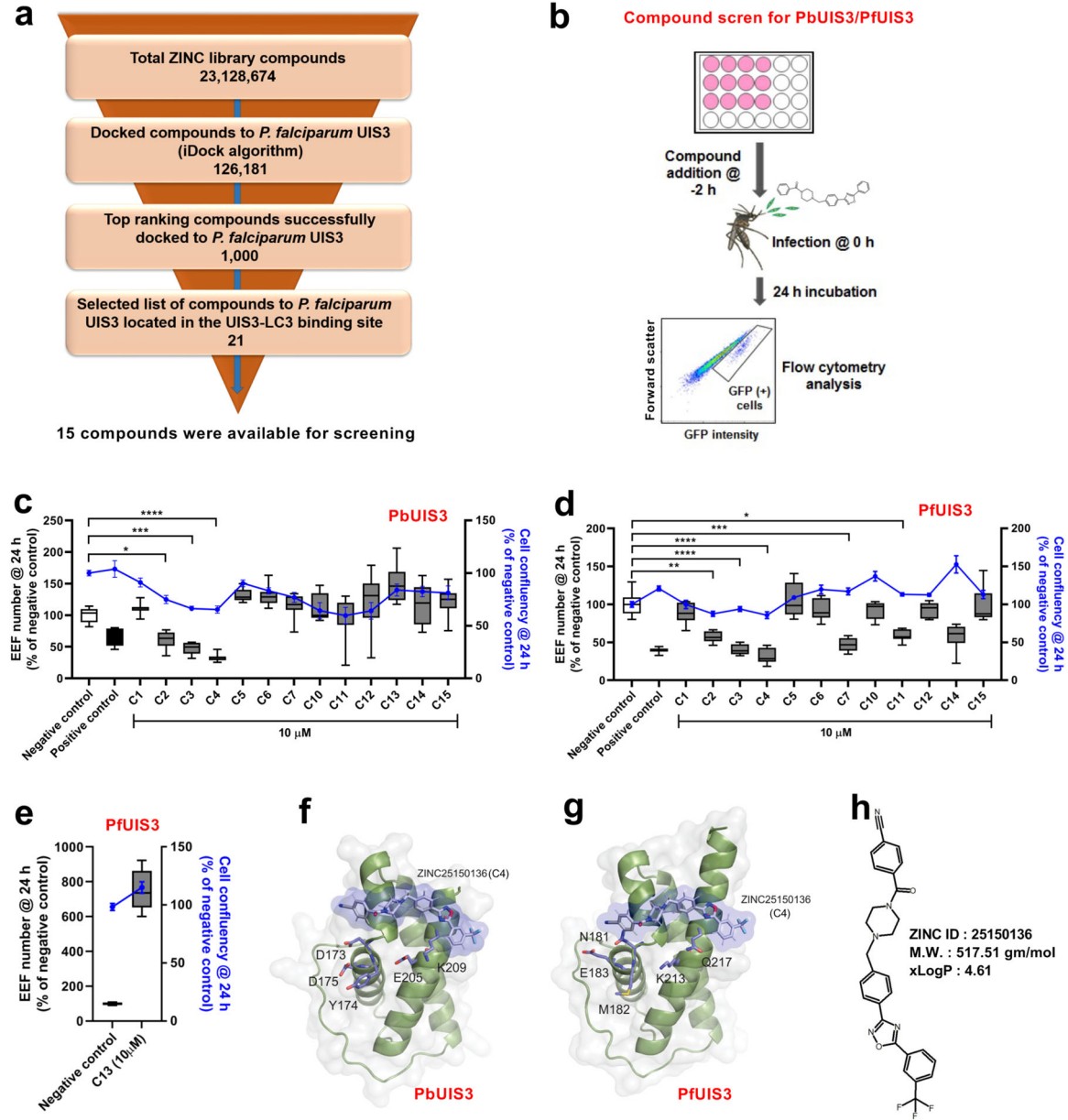

**Fig. 1 Virtual library screen (VLS) and phenotypic screen (PHS) of the selected hits for activity against *Plasmodium* liver-stages. a** Schematic representation of the different steps involved in VLS using ZINC database for *P. falciparum* with the number of the compounds identified in each phase. **b** Schematic representation of the experimental steps involved in PHS of selected compounds against *P. berghei* or PfUIS3@Pb infection in Huh7 cells. **c–e** Measurement of the antimalarial effect of the selected compounds in *P. berghei* (**c**) and PfUIS3@Pb (**d** and **e**) infected Huh7 cells using flow cytometry. For **c–e**, cells were infected 2 h after compound addition and analysed 24 h post-infection. The solvent of the compounds, DMSO (0.001%), was used as the negative control, and Primaquine (10 µM) was used as the positive control. Here, box-plots (10–90 percentile) represent the number of exo-erythrocytic forms, EEFs, as quantified by the number of infected cells. The blue lines (means ± SEM) represent cell confluency. C4 was identified as the most effective compound in case of both *P. berghei* and PfUIS3@Pb infection. $n = 3$ independent experiments. *P* values were calculated using one-way ANOVA with Tukey test, \**P* < 0.1, \*\**P* < 0.01, \*\*\**P* < 0.001, \*\*\*\**P* < 0.0001. **f, g** Ribbon and solvent accessible surface representation of the homology model of PbUIS3 (**f**) and the crystal structure of PfUIS3 (PDB code: 2VWA) (**g**) soluble domains (in grey) in complex with docked compound C4 (in blue). PbUIS3 homology model was built using the crystal structure of PfUIS3 as reference model. Chemical structure of compound C4 and amino acid residues involved in the interaction of UIS3 proteins with LC3 in *P. berghei* and *P. falciparum* are depicted in sticks. **h** Chemical structure and properties of C4.

a tri-fluoro-methyl-benzene and an N-alkyl-piperazine, linked to a nitrile derivative of benzoic acid (Fig. 1h).

**C4 impacts parasite survival inside the host cells.** Next, we sought to determine the optimal working concentration of C4 that can substantially decrease infection with both *P. berghei* ANKA and PfUIS3@Pb parasite lines. C4 decreased infection in a dose-dependent manner in both parasite lines (Fig. 2a, b; IC$_{50}$ @ 24 h

post-infection – 176.3 nM and 121.9 nM for *P. berghei* ANKA and PfUIS3@Pb lines, respectively). In the following experiments, where C4 was used at a concentration of 1 µM, our data clearly show that C4 decreases infection from a very early time point post-sporozoite addition. This effect increases with longer incubations up to 24 h post-infection and is maintained even at a later time point until 48 h post-infection (Fig. 2c, d and Supplementary Fig. 2). Importantly, C4 does not affect *Plasmodium* sporozoite

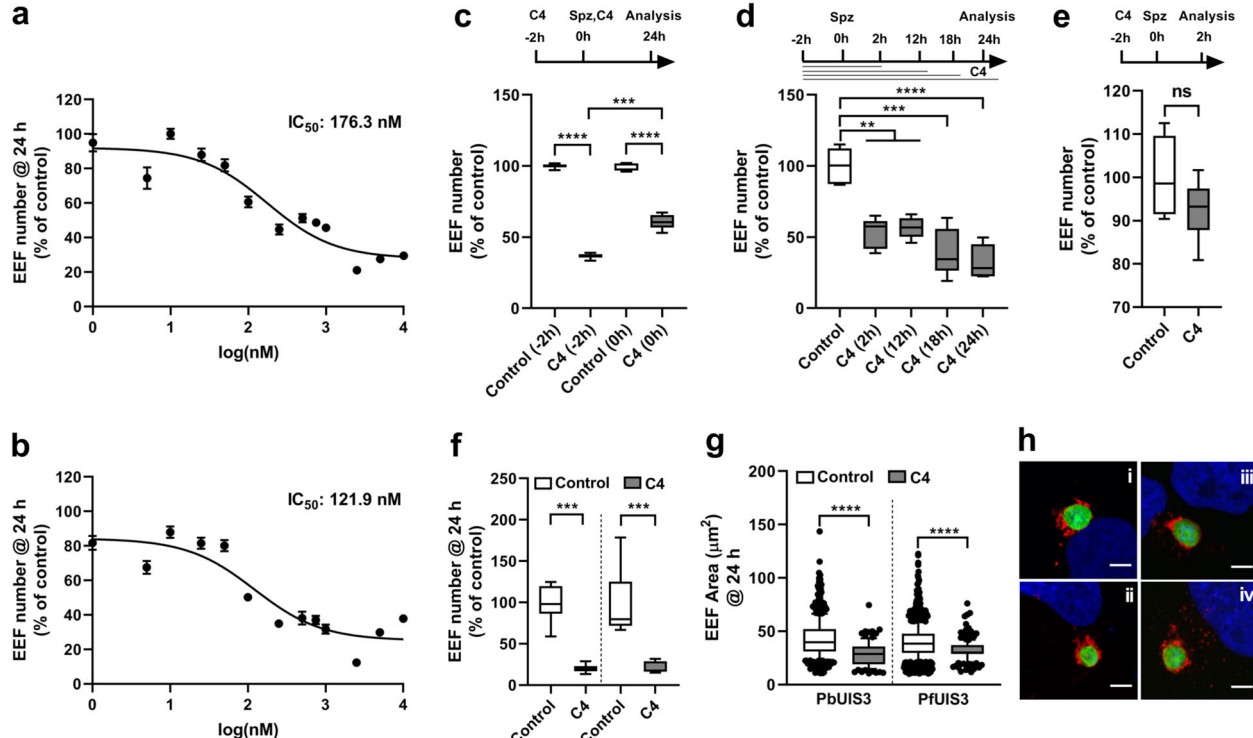

**Fig. 2 C4 affects parasite number, but not the invasion, during *Plasmodium* liver-stage infection. a, b** Dose-dependent effect of C4 on the number of EEFs of *P. berghei* (**a**) and PfUIS3@Pb (**b**) parasites in Huh7 cells. The effects were measured by flow cytometry at 24 h post-infection. Three-parameter curve fitting was carried out using GraphPad Prism. **c** Effect of 1 μM C4 on *P. berghei* infection in Huh7 cells. C4 was added to the cells 2 h before infection or at the time of infection. **d** Effect of 1 μM C4 treatment for various time intervals on *P. berghei* infection in Huh7 cells. C4 was added to the cells 2 h before infection. For (**c**) and (**d**) infection was measured by quantifying the EEFs number using flow cytometry at 24 h post-sporozoite addition. **e** Effect of 1 μM C4 on *P. berghei* invasion in Huh7 cells as quantified by EEFs number at 2 h post-infection using flow cytometry. Here, C4 was added to the cells 2 h before infection. **f** and **g** Immunofluorescence based quantification of the effect of 1 μM C4 treatment (added to the cells 2 h before infection) on EEFs number (**f**) and EEFs size (**g**) in *P. berghei* and PfUIS3@Pb infected Huh7 cells. For all the experiments, DMSO (0.0001%) was used as the negative control. In (**a**) and (**b**), data represents means ± SEM, $n = 3$ independent experiments. In (**c**–**g**), data is represented as box-plot (10–90 percentile), $n = 3$ independent experiments. For (**c**, **d**, **f**) and (**g**), P-values were calculated using one-way ANOVA with Tukey test, $**P < 0.01$, $***P < 0.001$, $****P < 0.0001$. For (**e**), non-parametric two-tailed Mann-Whitney test was used to compare the effect of DMSO control and C4. ns: non-significant. **h** Representative confocal images of EEFs size comparison (i & ii: *P. berghei* infection, control and C4; iii & iv: PfUIS3@Pb infection, control and C4). Cells were stained with anti-UIS4 (red), anti-GFP (green) and Hoechst (blue). Scale bars = 5 μm.

invasion of host cells (Fig. 2e). Microscopic analysis of both *P. berghei* ANKA and PfUIS3@Pb parasite-infected Huh7 cells confirms that C4 impacts parasite infection through a significant decrease on the number of infected cells (Fig. 2f), albeit also having a modest impact on parasite development (measured by parasite area) (Fig. 2g, h). Together, these data imply that C4 reduces *Plasmodium* infection by interfering with parasite survival during development inside the host cell.

**C4 interacts with the UIS3-LC3 complex.** To determine whether the detrimental effect of C4 on *Plasmodium* survival was due to its impact on the UIS3-LC3 interaction, we performed Isothermal Calorimetry (ITC) using purified recombinant UIS3 and LC3 proteins. We confirmed the direct binding of UIS3 and LC3 with ITC (Fig. 3a). Adding C4 to the complex showed strong interaction, with a 10-fold higher affinity to the UIS3-LC3 complex, in comparison to the direct interaction of the two proteins (Fig. 3a, b; $K_d$- 2.15 ± 0.121 μM for UIS3-LC3 direct binding, $K_d$- 0.241 ± 0.0011 μM for C4 binding to the UIS3-LC3 complex). This change in $K_d$ value indicates that C4 binds to the UIS3-LC3 complex and can compete for the UIS3-LC3 interaction.

**The anti-parasitic activity of C4 depends on host cell autophagy.** We have previously shown that UIS3 interacts with host

LC3 to avoid parasite elimination by autophagy in hepatocytes. Having now found that C4 indeed binds to the complex with high affinity, we hypothesised that this could interfere with the complex function and decrease *Plasmodium* infection by interfering with the parasite ability to evade host cell autophagy. To test this, we have assessed the anti-parasitic activity of C4 in both autophagy-competent and autophagy-deficient cells, using both genetic and chemicals tools. The results show that C4 only reduces *Plasmodium* infection in cells with a functional autophagy pathway. Indeed, C4 does not cause any reduction in infection in Atg5- and LC3-depleted host cells (Fig. 4a, b, Supplementary Fig. 3 and Supplementary Fig. 8). The impact on infection in C4-treated control cells (scrambled siRNA) is comparable to that in Huh7 cells (Fig. 4a, b). Moreover, treatment of Huh7 cells with chloroquine (a known autophagy flux inhibitor) rescues C4 effect on the number of infected hepatocytes (Fig. 4c, d). Finally, LAMP1 staining throughout infection show that lysosome dynamics around the PVM is distinct in the presence or absence of C4 (Fig. 5a, b). Altogether, these results demonstrate that C4 impacts infection via the host cell autophagy machinery.

**C4 does not interfere with the intrinsic host autophagy pathway.** Any effective anti-malarial strategy must impede infection without interfering with intrinsic host functions. As C4 acts on a

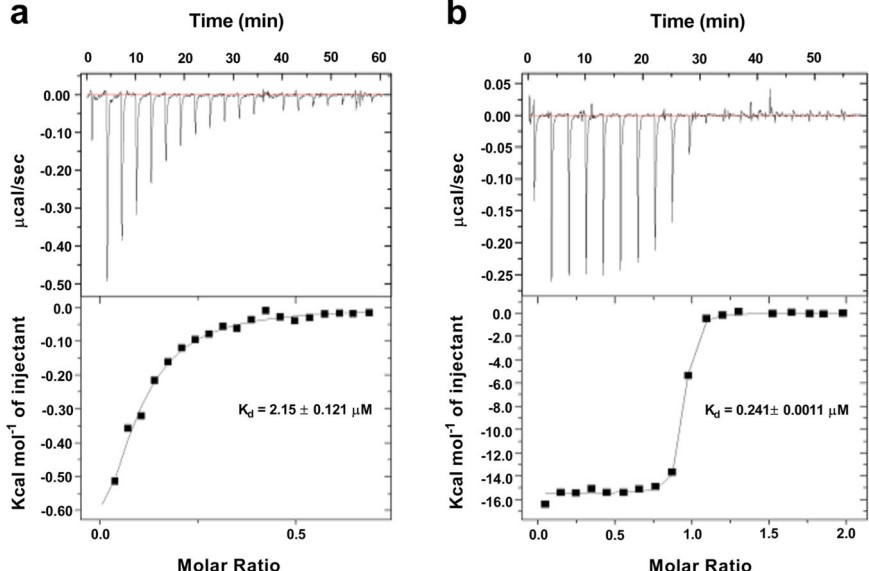

**Fig. 3 C4 interacts with the UIS3-LC3 complex.** ITC (Isothermal calorimetry) performed using C4, PfUIS and LC3 showing that C4 binds to the UIS3-LC3 complex. **a** The raw ITC curve and the ΔH from integration for titration of 100 µM LC3 into 10 µM PfUIS3. **b** The raw ITC curve and the ΔH from integration for titration of 100 µM C4 into a saturation solution of UIS3 with LC3. Both (**a** and **b**) fits a one-site binding model and their dissociation constant of the interaction, $K_d$, is shown. $n = 3$ independent experiments. Representative data is shown.

protein-protein interaction at the host-parasite interface, we next sought to determine the effect of C4 administration on host cell normal autophagy activity. To that end, we induce autophagy in HeLa cells in the presence or absence of C4. The results clearly show that C4 does not cause any impairment in the host cells' ability to efficiently respond to an exogenous autophagy stimulus (Fig. 6a, b and Supplementary Fig. 7).

Altogether, our data establish an efficient anti-malarial strategy (Fig. 7) by proving that the disruption of a critical host-parasite interaction without affecting the normal host function strongly impacts the first and obligatory step of *Plasmodium* parasites infection.

## Discussion

For intracellular pathogens, the host cell acts as the replicative compartment that provides the pathogen with the necessary resources such as nutrients and, energy, required for their development as well as shields them from the components of the host immune system such as complements. However, host cells are equipped with various intracellular defence mechanism(s) that can destroy the invading pathogen. Autophagy is emerging as one of the most remarkable tools of the intracellular defence weaponry that pathogens must confront upon host cell invasion.

Intracellular organisms have evolved various strategies to subvert the host autophagy response and manipulate this process at the molecular level to establish a successful infection inside the host cell. The intracellular bacteria *Shigella* residing in its vacuolar compartment only for a short duration secretes an effector protein that surrounds the bacteria while in the host cell cytosol and impairs their detection by the host autophagy machinery[16]. However, other bacteria such as *Coxiella burnetii* replicates in an autophagosome-like vacuolar compartment by delaying lysosomal fusion with it[17]. Several viruses such as *Herpes Simplex* block the initiation of autophagy flux by modulating host signalling pathway[18]. In the case of the apicomplexan *Toxoplasma gondii*, the host autophagy response is inhibited by two microneme proteins, MIC3 and MIC6, which activate the host EGFR/Akt

signalling pathway and thereby protect its vacuolar compartment from fusing with the lysosome[19].

During the obligatory liver infection, which allows the first clonal expansion of the malaria parasite *Plasmodium* in the mammalian host, autophagy plays seemingly opposing roles that greatly impact on parasite survival and development within the host cell. *Plasmodium* hepatic stage induces a canonical autophagy response, which might act as a possible nutrient source supporting the parasite's maturation into the disease-causing merozoites[12,20] as rapamycin treatment or starvation increases the number of liver-stage parasites[21]. Simultaneously, host cell triggers a non-canonical autophagy response called PAAR (*Plasmodium* associated autophagy-like response) against the parasite leading to the association of autophagy markers such as LC3, P62, NBR1, NDP52 and ubiquitin with the *Plasmodium* harbouring PVM[13,20,22]. Therefore, for parasites to complete the liver-stage of infection, they must develop a strategy to circumvent the host's autophagy defence response without impairing salvage of the autophagy-derived metabolites.

*Plasmodium* PVM resident proteins are at the right place to play functional roles that ensure the development of the parasites inside hepatocytes. During liver infection, *Plasmodium* develops surrounded by a PVM that shields the parasite from host cytosolic defence mechanisms. Although originally derived from the host cell plasma membrane, *Plasmodium* extensively modifies PVM's composition by inserting parasite-derived proteins into it[23–25]. Until now only a few liver-stage PVM proteins have been identified whose molecular function is known. UIS3 is one of them that is essential for *Plasmodium* liver-stage development as a mutation of this protein results in parasite elimination at early stages of infection[26]. While it has been proposed that LC3 dissociation from PVM at later stages of infection (40 h) is necessary for *Plasmodium* intrahepatic development[15,20] our recent work has proved that the parasite protein UIS3 binds to, and retains, LC3 on the PVM from a very early stage of infection, which protects the parasite from host autophagy machinery and supports parasite survival and development within the hepatocytes[11]. Previously, we have presented several pieces of evidence that

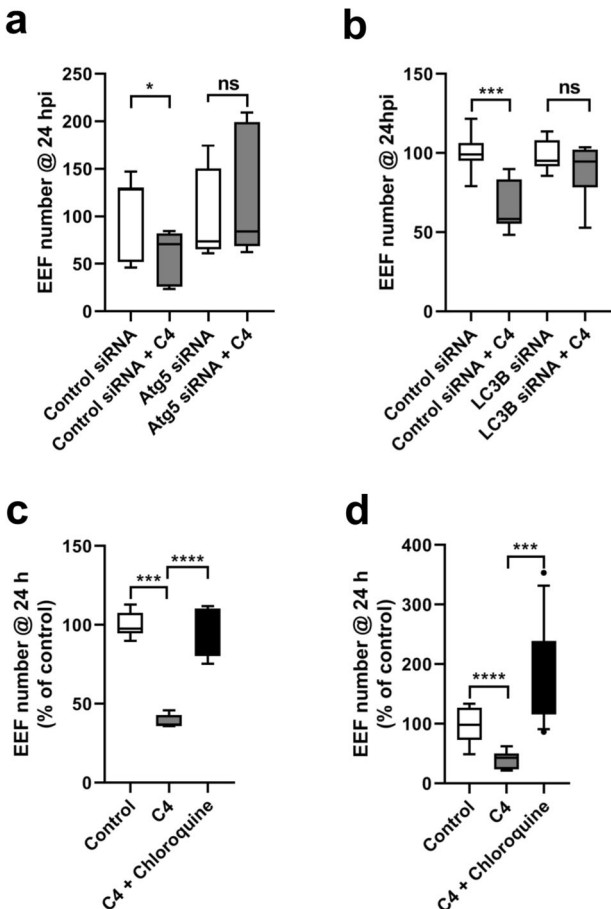

**Fig. 4 C4 employs host autophagy pathway for its anti-plasmodial effect leading to parasite elimination from the host cells. a, b** Effect of 10 μM C4 on *P. berghei* infection in control (WT HeLa) and autophagy-deficient (Atg5 knockdown in 4a; LC3B knockdown in 4b) HeLa cells. Here data is represented as box-plot (10–90 percentile), n = 3 independent experiments. Statistical significance was assessed using non-parametric two-tailed Mann-Whitney test. ns: non-significant, *P < 0.1, ***P < 0.001. The anti-parasitic effect of 1 μM C4 on *P. berghei* (**c**) and PfUIS3@Pb (**d**) infection in Huh7 cells can be reverted using the known autophagy inhibitor Chloroquine (50 μM). Here, C4 and Chloroquine were added to the cells 2 h before infection. The effect was analysed at 24 h post-infection using flow cytometry by quantifying the EEFs number in infected cells. Data is represented as box-plot (10–90 percentile). n = 3 (for 4c) and n = 4 (for 4d) independent experiments. Statistical significance was assessed using the unpaired two-tailed *t*-test. ***P < 0.001, ****P < 0.0001.

confirmed the protective function of UIS3 during *Plasmodium's* intrahepatic development. While *uis3*-deficient parasites cannot develop in wild type hepatocytes, they develop inside autophagy-deficient host cells which strongly suggests that UIS3 plays a critical role in the interaction between the parasite and the host autophagy pathway. Using exogenously expressed UIS3 in HeLa cells, we showed that UIS3 directly binds to LC3, possibly through its non-canonical LIR motif. Furthermore, by molecular docking analysis and in vitro experiments, the residues on UIS3 potentially involved in this interaction have been identified[11]. Together, these data indicate that by sequestering LC3 on the PVM, UIS3 blocks its binding to other autophagy target proteins such as P62 and Rab7 effector proteins leading to the inhibition of host autophagy defence response[27,28].

Considering the protective role of UIS3 as well as the specificity of the UIS3-LC3 interaction, we sought to identify small molecule

inhibitors of this protein-protein interaction that can impair *Plasmodium's* liver infection. First, by molecular docking analysis using an open-access compound library, we identified a list of candidates with the putative potential to disrupt UIS3-LC3 interaction. We next conducted a PHS using *Plasmodium* in vitro infection models and selected the best hit, C4, that substantially decreased *Plasmodium's* liver infection levels, mostly by affecting the parasite survival within the hepatocytes. By Isothermal calorimetry assay, we confirmed that C4 directly binds to the UIS3-LC3 complex with high affinity and could act by interfering UIS3 binding with LC3. Using genetic and chemical tools that can modulate host cells autophagy pathway, we showed that C4 employs host autophagy machinery to exert its anti-parasitic activity. Interestingly, C4's activity does not affect the ability of the host cell to induce a normal autophagy flux in response to an appropriate stimulus. Based on these four distinct sets of data – namely molecular docking analysis, isothermal calorimetry assay definitively showing C4 capability of binding UIS3-LC3 complex, the lack of effect of C4 in autophagy-deficient host cells and finally the distinct lysosome dynamics around the PVM in the presence or absence of C4 – we propose that C4 mode of action as an anti-malarial compound relies on its ability to disrupt UIS3-LC3 interaction inhibiting the parasite's ability to evade the host autophagy response and causing parasite elimination by the host cell (Fig. 7). Lysosome association with the PVM is a fascinating – possibly necessary[14,20] but conceivably also deleterious in certain conditions or above a certain level – aspect of host-*Plasmodium* interactions during the liver-stage of infection. It is important to note that even at the highest concentration of C4 tested, some parasites seem to survive C4 treatment. In agreement with this observation, it has been previously shown that infection with *uis3*-deficient parasites show frequent breakthroughs[29]. Indeed, not all wild type cells present the same autophagy capacities and (i) we have previously shown that *uis3*-deficient parasites fully develop in autophagy-deficient cells[11] and, (ii) we now show that C4 is only active in autophagy-competent cells.

Given its fundamental role in cellular homoeostasis and during cellular stress conditions, autophagy plays a crucial role in various diseases such as cancer, cardiovascular disease, autoimmune disease, neurodegenerative disease, diabetes, obesity, tuberculosis and leprosy infection and chronic inflammatory disease[30]. Different phases of this process, including autophagy signalling and, autophagosome formation and maturation, represent valid targets for the development of novel drugs with therapeutic potential[30]. Our data, to the best of our knowledge, validate for the first time the druggability of a critical *Plasmodium*-host protein-protein interaction. This information provides the starting point for developing an autophagy modulator that can be used as an efficient antimalarial prophylactic for malaria liver infection. Notably, our strategy targets the malaria parasite explicitly without interfering with the host autophagy machinery, critical for several physiological functions of our body, thus avoiding adverse side effects in the treated individuals. The drug developed based on this strategy will have the potential to stop infection at an early stage and avoid not only the blood-stage pathology but also blocking disease transmission[31]. Such a drug would enjoy a tactical advantage over currently available drugs against the blood-stage of infection, given that the reduced number of parasites during liver infection should delay the development of resistance. Furthermore, being a liver-stage targeting drug that acts by employing the host autophagy machinery, it may also have activity against the *P. vivax* dormant form (hypnozoites), which will prevent relapsing malaria[32] and thereby contribute to any malaria eradication campaign. Currently, primaquine and tafenoquine are the only approved drugs with known efficacy against the liver-stage. Among these, primaquine has liabilities, and its

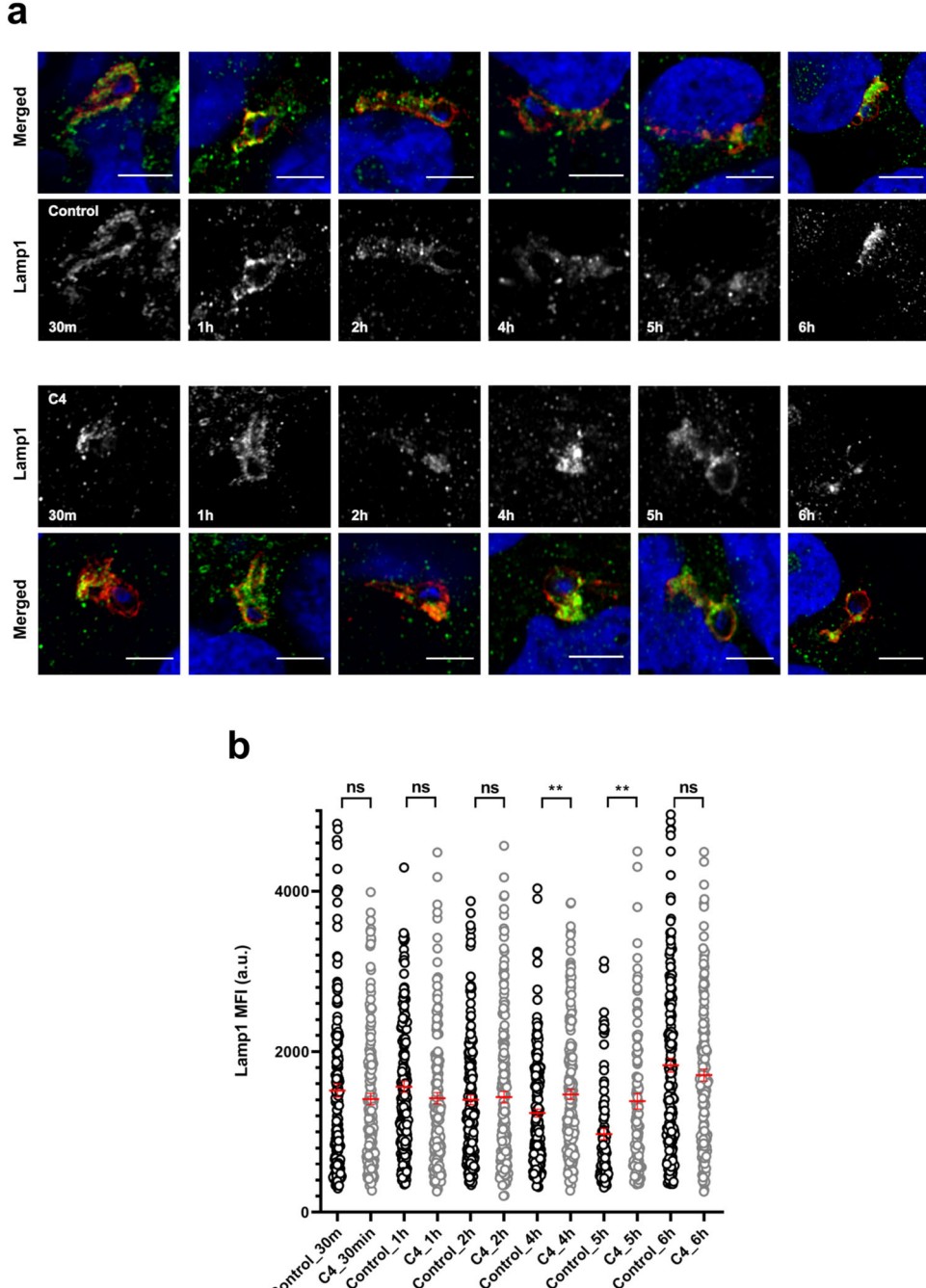

**Fig. 5 C4 increases lysosomal marker protein Lamp1 colocalization with the liver-stage PVM.** Huh7 cells infected with *P. berghei* parasite were treated with 1 μM C4 and fixed at different time point post-compound addition. DMSO (0.0001%) was used as the negative control. **a** Representative confocal images of *P.berghei* infected Huh7 cells treated with C4/DMSO. Cells were stained with anti-UIS4 (red), anti- Lamp1 (green) and Hoechst (blue). Scale bars = 5 μm. **b** Quantification of Lamp1 intensity around the PVM. The graph represents Lamp1 mean fluorescence intensity (MFI) around the PVM marked by UIS4. The data represents means ± SEM (*n* = 2 independent experiments). N ≥ 100 parasites. *P* values were calculated using unpaired two-tailed *t*-test. ns: non-significant, **$P < 0.01$.

ability to cause haemolytic anaemia in patients with glucose 6-phosphate dehydrogenase deficiency has severely restricted its use[33]. On the other hand, although tafenoquine is less prone to produce oxidative stress, it is still not advised to a major fraction of the human population. Indeed, both drugs are not allowed for pregnant women and children[34].

Overall, our data provide the proof-of-concept for the prophylactic targeting the UIS3-LC3 interaction during *Plasmodium* liver infections and the use of UIS3-LC3 interaction inhibitors that can substantially impair the first obligatory step of malaria infection in the human host alone or through combinatorial treatments with existing antimalarials. However, in order to develop such an inhibitor with suitable bioavailability and pharmacokinetic properties, direct structural information on the UIS3-LC3 interface will be essential. This strategy of disrupting an autophagy-dependent, host-parasite interaction without

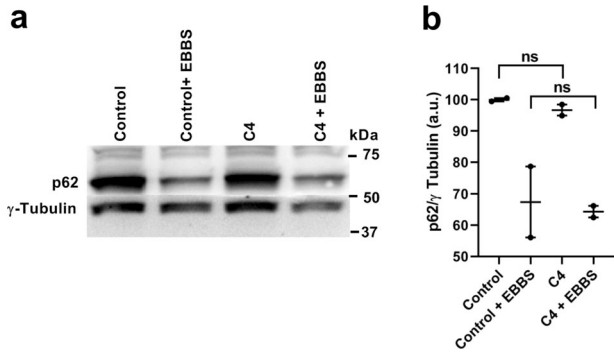

**Fig. 6 C4 does not obstruct the intrinsic autophagy flux of the host cell.** **a** Representative immunoblot ($n = 2$ independent experiments) of p62 levels in HeLa cells that were kept in complete (control) and amino acid depleted medium (amino acid starvation) for 7 h. **b** The p62 levels before and after amino acid starvation were calculated as the ratio of p62 to γ tubulin (loading control). Data represents means ± SEM ($n = 2$ independent experiments). Statistical significance was assessed using the unpaired two-tailed t-test. ns: non-significant.

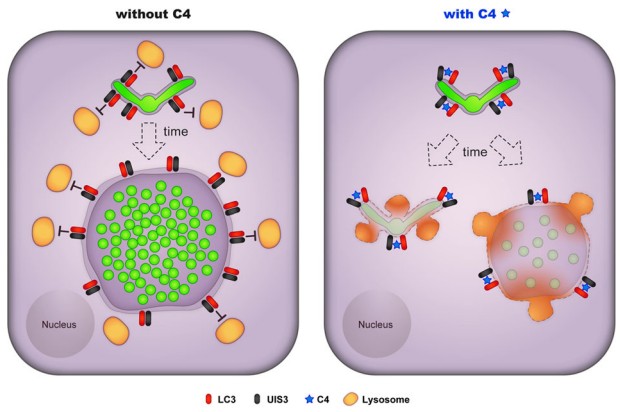

**Fig. 7 Proposed mode of action for the anti-malarial small molecule C4 on *Plasmodium* liver-stage infection.** The data supports the model that C4 overcomes the protective function of UIS3 during *Plasmodium* intrahepatic development. In the presence of C4, UIS3 cannot sequester LC3 rendering the downstream autophagy receptors to interact with it. As a result, the host autophagy pathway remains active leading to the lysosome fusion with the PVM and elimination of the parasite from the hepatocyte.

affecting the host's normal autophagy response can be extended to other intracellular pathogens that maintain similar autophagy-centric relationships with their hosts.

## Methods
**Chemicals**. RPMI 1640, DMEM, EBSS and other cell culture reagents were purchased from Gibco Invitrogen. All chemicals were from Sigma-Aldrich, unless specified otherwise. Reagents for molecular cloning (restriction enzymes, ligase, buffers) were purchased from New England Biolabs. siRNAs were transfected using Lipofectamine RNAiMAX (Thermo Fisher Scientific) in OptiMEM (Invitrogen).

**Cell lines**. Huh7 and HeLa (ATCC) cells were cultured under standard conditions in RPMI 1640 and DMEM medium respectively supplemented with 10% FCS, 1% glutamine, 1% non-essential amino acids (RPMI 1640), 1% penicillin/streptomycin, and 1% Hepes (RPMI 1640). All cell lines were routinely tested for mycoplasma contamination using a luminescence-based mycoplasma detection kit (Lonza).

**Mice**. All the mice used in this study were C57BL/6 J males, of age between 6 and 8 weeks and, housed in the animal facility of the Instituto de Medicina Molecular, Lisbon. All the protocols were approved by the internal animal care committee of the institute and were performed according to national and European ethical regulations.

**Parasite lines**. *P. berghei* sporozoites were obtained through dissection of the salivary glands of infected female *Anopheles stephensi* mosquitoes bred at the Instituto de Medicina Molecular. The following parasite lines were used: GFP and firefly luciferase–expressing *P. berghei* ANKA (676m1cl1), GFP expressing *P. berghei* ANKA (259cl2), GFP expressing (resistance marker free) *P. berghei* ANKA (507cl1) [all three from Leiden Malaria Research Group], and *P. falciparum UIS3* tagged with HA as well as GFP expressing *P. berghei* ANKA (PfUIS3@Pb, generated for this study).

***Plasmodium* transfection**. PfUIS3@Pb parasites were obtained by replacing the *P. berghei uis3* open reading frame (ORF) with *uis3* ORF of *P. falciparum* via double crossover recombination as described in Supplementary Fig. 1a. Transfection was carried out by electroporation of purified schizonts[35]. Parasites were maintained under Pyrimethamine selection pressure, collected on day 10 post-transfection and genotyped by PCR with the primers listed in Supplementary Table 3.

**RNAi**. HeLa cells were transfected with LC3B siRNA (Cell Signalling Technology, 6213, 100 nM) or Atg5 siRNA (Cell Signalling Technology, 6345, 100 nM) using Lipofectamine RNAiMAX as per manufacturer protocol. The efficiency of knock-down was assessed at 24 h (for LC3) and 48 h (for Atg5) after transfection by western blot. The following primary antibodies were used: LC3 (rabbit polyclonal, MBL, PM036, 1:1000), Atg5 (rabbit polyclonal, Cell Signalling Technology, 2630, 1:1000) and β actin (mouse monoclonal, Abcam, ab8224, 1:1000). The expression level of the target genes was calculated as the ratio of target gene to β actin (loading control).

**Virtual ligand screening and molecular docking of PfUIS3-ligand complex**. Structure of the molecular models of UIS3-LC3 complex was previously described[11]. The screening of inhibitors of UIS3-LC3 binding was performed by the iDock algorithm[36] and the ZINC chemical compound database. A cubic searching space of 30 ×30 x 30 Å centred at the LC3-binding pockets located in the surface of UIS3 protein from *P. berghei* and *P. falciparum* were used as a starting point for the virtual screening. Among the 23,128,674 ligands in ZINC database, 126,181 were successfully docked to UIS3 surface pockets. From the 1,000 top ranked ligands sorted by the Gibb's function, the best 21 docking solutions involving the UIS3 residues previously characterised as important for LC3 binding[11], were further refined by energy minimisation using a knowledge-based potential of mean force and stereochemistry correction with GROMACS[37].

**Luciferase assay**. Huh7 cells were seeded on a 96 well plate (10,000 cells/well) in RPMI media. The next day, media was replaced with DMSO (0.0001%)/1 μM C4 containing media. After 2 h, cells were infected with luciferase–expressing *P. berghei* sporozoites (10,000/well) and processed for Luciferase assay[38] at 48 h post-infection.

**Flow cytometry analysis of liver-stage infection**. Huh7 cells were seeded on a 24 well plate (50,000 cells/well) in RPMI media respectively. The next day, media was replaced with different compounds (10 μM)/ C4 (various conc.)/1 μM C4/1 μM C4 + 50 μM Chloroquine containing media, as appropriate in different experiments. After 2 h cells were infected with *P. bergehi*/PfUIS3@Pb parasites and analysed by flow cytometry at 2 h/24 h post-infection. DMSO was used as negative control and Primaquine was used as the positive control. siRNA transfected HeLa cells (30,000 cells/well) were treated with 10 μM C4/DMSO (0.001%) as above. At 24 h (for LC3) and 48 h (for Atg5) after transfection the cells were infected with *P. berghei* parasite and analysed by flow cytometry at 24 h post-infection. For flow cytometry, infected cells were trypsinized and resuspended in 300 μl of media. 50 μl of each sample were analysed on BD Accuri C6 as before[39]. Data were analysed using FlowJo.

**Immunofluorescence**. To assess C4 effect on EEF number and size, Huh7 cells were plated on glass coverslips (40,000/cover slip), treated with 1 μM C4 or 0.0001% DMSO (control) and after 2 h infected with *P. berghei*/PfUIS3@Pb sporozoites (40,000/cover slip). To characterise PfUIS3@Pb parasite line, Huh7 cells were seeded on cover slips as above and infected with WT (*P. berghei* ANKA (507cl1)/PfUIS3@Pb sporozoites (40,000/cover slip). 24 h after, cells were fixed in 4% paraformaldehyde (ChemCruz) for 15 min at room temperature (RT), permeabilized/blocked (PBS, 0.1% Triton X-100, 1% BSA) for 1 h at RT and incubated with primary antibodies (diluted in blocking solution) for overnight at 4 °C. Cells were then washed with PBS, incubated with AlexaFluor-conjugated secondary antibodies (Invitrogen) and Hoechst 33342 (Invitrogen) for 1 h at RT and washed again. The coverslips were mounted with Fluoromount (SouthernBiotech). To analyse Lamp1 dynamics around the PVM, Huh7 cells were plated on glass coverslips (50,000/cover slip) and infected with *P. berghei* sporozoites (40,000/cover slip) on the following day. after 3.5 h post-infection, cells were treated with 1 μM C4 or 0.0001% DMSO (control) and fixed at different time points post-compound addition using 4% paraformaldehyde as above. Fixed cells were permeabilized (0.1% Triton X-100 in PBS) for 5 min at RT, washed with PBS and blocked (3% BSA in PBS) for 20 min at RT. After that the cells were incubated in primary and secondary antibodies as above. The coverslips were mounted with Fluoromount. The following primary antibodies were used: PbUIS4 (goat polyclonal,

SicGen, AB0042-200, 1:1000), HA (mouse monoclonal, BioLegend, 901509, 1:500) and Lamp1 (rabbit polyclonal, Sigma, L1418, 1:1000). GFP signal was detected using AlexaFluor 488 conjugated anti GFP (rabbit polyclonal, Invitrogen, A-21311, 1:500) antibody.

For infection quantification by microscopy, images (36 per cover slip) were acquired on a Zeiss Axiovert 200 M wide-field microscope equipped with an automated stage. Images to quantify Lamp1 intensity dynamics were acquired on Zeiss Axio observer with a 506 mono-CCD Axiocam at 40x magnification (40x/ 0.75 M27, Zeiss). High-resolution images were acquired on Zeiss Laser scanning confocal microscopes (LSM 710/LSM 880) at 63x magnification (63x/1.4 Oil DIC M27, Zeiss). All images were processed and analysed using FIJI.

**Protein purification**. His-mycUIS3 and GST-His LC3 were expressed in BL21 (DE3) pLysS cells, cultured at 37 °C and the protein was induced with 0.3 mM IPTG at an $OD_{600}$ of 0.8. After IPTG induction the cells were cultured at 18 °C for 16 h. The cells were pelleted and resuspended in lysis buffer containing 50 mM Sodium Phosphate at pH 8, 300 mM NaCl, 10 mM Imidazole, 2 mM PhenylMethylSulphonyl Fluoride (PMSF) and Protease inhibitor cocktail (Roche). The proteins were purified by passing the lysate through His-trap Ni-charged columns (GE healthcare). The eluted proteins were proteolytically cleaved overnight with thrombin for myc-UIS3 and rTEV for LC3 at 4 °C to remove the tags (His tag from His-myc-UIS3 and GST-His tag from GST-His LC3). The proteins were then passed through a superdex-75 PG column (GE Healthcare). The proteins were finally concentrated to 10 mg/ml concentration in a final buffer containing 50 mM Sodium Phosphate and 150 mM Sodium Chloride at pH 6.5.

**Isothermal calorimetry titrations**. ITC experiments were performed using MicroCal-ITC 200 (GE healthcare) at 25 °C and the data was analysed with MicroCal ITC-origin analysis software based on one-site binding reaction. For direct interaction studies, the PfUIS3 was taken at 10 μM concentration and titrated against 0.100 μM of LC3. For competition assay, a saturated solution of UIS3 with LC3 was titrated against 100 μM of C4. C4 was resuspended in the same final buffer (with 2% DMSO) in which UIS3 and LC3 was dissolved.

**Autophagy induction and p62 degradation assay**. HeLa cells treated with 1 μM C4 or 0.0001% DMSO (control) for 24 h were changed from normal growth medium to EBBS to induce amino-acid starvation dependent autophagy. After 7 h, cells were collected in lysis buffer (50 mM NaCl, 50 mM Tris-Cl pH 8, 5 mM EDTA pH 8, 1% Triton X-100, protease inhibitor) and analysed by western blot using the following antibodies: p62 (rabbit polyclonal, Sigma-Aldrich, P0067, 1:1000) and gamma-tubulin (mouse monoclonal, Sigma-Aldrich, T5326, 1: 10,000). p62 levels were measured by quantifying the ratio of p62 to tubulin signals using Image Lab.

**Phenotypic characterisation of the PfUIS3@Pb**. PVM localisation of PfUIS3-HA and liver-stage development of PfUIS3@Pb parasites (EEF number and size comparison with WT) was done by immunofluorescense microscopy as described before.

In vivo infection progression and development of severe pathology was assessed in C57Bl/6 males by injecting a total of 2500 WT (P. berghei ANKA (507cl1) or PfUIS3@Pb sporozoites (5 mice/group). Parasitaemia (% parasite in mouse blood) and survival was monitored until day 16 post infection. For parasitaemia, 5 μl blood from each mouse was diluted in 200 μl PBS and was analysed by BD LSRFortessa.

**Statistics and reproducibility**. Data represents means ± SEM. Statistically significant differences between two different groups were determined using One-way ANOVA with Tukey test, Non-parametric two-tailed Mann-Whitney test or Unpaired two-tailed t-test, as indicated in the figure legends. Statistical significances are represented in the figures as follows: ns: non-significant, $*P < 0.1$, $**P < 0.01$, $***P < 0.001$, $****P < 0.0001$. The number of biological replicates and sample size are described in the respective figure legends. All the tests were carried out in GraphPad Prism. Samples in the experiment for Lamp1 dynamics analysis were randomised. Sample sizes on mice experiments were chosen based on historical data. No statistical methods were used to predetermine sample size.

**Reporting summary**. Further information on research design is available in the Nature Research Reporting Summary linked to this article.

## Data availability

All data generated or analysed during this study are included in this manuscript and its Supplementary Information. Source data underlying the graphs and charts presented in the main figures are available as Supplementary Data 1. Full blots are shown in Supplementary Information.

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

## Acknowledgements

We thank Ana Parreira for producing *Plasmodium*-infected *Anopheles* mosquitoes and Ksenija Slavic for helping with transgenic *Plasmodium* line development. We also thank Vanessa Zuzarte-Luís for essential suggestions, João Mello-Vieira for help with immunofluorescence experiments and Mahavir Singh for ITC assays. This work was supported by grants from Institut Mérieux (MRG_20052016 to M.M.M). S.S. and A.F.C. were recipients of Fundação para a Ciência e Tecnologia fellowships SFRH/BPD/116451/2016 and SFRH/BPD/112009/2015, respectively. H.R. and V.S. were supported by core funds from NCBS-TIFR. A.L. was supported by Sanofi-Institut Pasteur 2018 Prize to M.M.M.

## Author contributions

S.S. and M.M.M. conceived and led the study, designed experiments and wrote the manuscript. S.S. performed experiments, acquired data, performed data analysis and interpretation. F.J.E. performed molecular docking and virtual compound library screen. A.F.C. and S.S. conducted animal experiments; A.F.C. performed data analysis and interpretation for these experiments. H.R. performed protein purification. H.R. and V.S. conducted Isothermal calorimetry analysis. A.L. performed the imaging and image analysis for lysosomal dynamics. S.M. helped with transgenic *Plasmodium* line development. All authors read and approved the final manuscript.

## Competing interests

The authors declare no competing interests.
