## [Peer Review File · Communications Biology]

Reviewers' comments:

Reviewer #1 (Remarks to the Author):

Setua and colleagues present an interesting screen for small molecules that interrupt the Plasmodium UIS3-host LC3 interaction. They performed an in-silico screening of more than 20 million compounds and identified 21 compounds that have the potential to bind the UIS3-LC3 interacting region. 15 compounds were tested in a phenotypic screen and one, termed C4, was found to block parasite development in vitro. Further in silico analysis confirmed C4 binding to a pocket that partially overlaps the UIS3-LC3 interacting region. C4 was found to significantly reduce parasite numbers in vitro. When host cell autophagy was blocked by genetic deletion of Atg5 or by chloroquine, the effect was rescued suggesting that C4-mediated elimination of parasites depends on host cell autophagy.

The results of this study are promising but preliminary in many respects.

Major concerns

In silico analysis revealed 21 compounds but, in the end, only 1 had a consistent effect on the different parasite strains tested. This shows the strength and the weakness of in silico screening at the same time. The strength is certainly that they were able to identify a potent compound. The weakness is that in most other cases the prediction has been wrong. My question is now how can the authors be sure that C4 really binds as they predict? Did they perform the in-silico docking experiments like presented for C4 in Figure 1f and g also for all other 14 compounds and did these fail to bind? In particular, according to their results presented in figure 1c and d, compound C2 should only bind to PbUIS3 and C3 only to PfUIS3. I find it also very strange that they compare luciferase assays with EEF numbers. They should perform comparable assays for both parasite strains that allow a proper comparison.

Their statement: "As predicted, C4 binds in a pocket that partially overlaps the proposed UIS3-LC3 interacting region for both *P. falciparum* and *P. berghei* UIS3 (Fig. 1f and g)." is clearly overinterpreted and it indeed needs UIS3-C4 co-crystallization experiments to confirm their assumption. It is very surprising that C4 does not directly interact with any of the key interacting amino acids identified earlier (Real et al. 2018, Nature Microbiology). Has any of the mutants described in the Real paper been tested for C4 interaction? They should extend their in-silico analysis and define mutations that do not allow C4 binding to either PfUIS3 or PbUIS3. These UIS3 mutations should then be generated and expressed recombinantly and tested for C4 binding.

They clearly need to show that C4 treatment indeed blocks LC3 labeling of the PVM. I suggest to treat GFP-LC3 expressing *P. berghei*-infected Huh7 cells with C4 and analyze loss of LC3 binding to the PVM.

They also should do IFAs to analyze the PVM after C4 treatment using common PVM marker proteins like UIS4 or Exp1. This should give a clear idea whether the PVM is compromised. Another very important experiment is to infect LC3-deficient cells and treat them with C4, which should not have an additional effect on parasite survival. It would also be very important to test whether UIS3 deficient parasites can normally develop in LC3-deficient cells.

A crucial experiment they describe is the infection and treatment of autophagy-deficient Atg5^{-/-} cells as this provides genetic indication that autophagy is eliminating intracellular parasites. However, MEF cells are very poor host cells for *P. berghei* as *P. berghei* does not develop normally in this type of cells. Given the importance of this experiment, they should use better suited Atg5-deficient host cells. They also need to show that in C4-treated WT cells, parasites end up in an autophagosome and are indeed eliminated by autophagy. In support of this notion, C4 has a rather modest effect on parasite elimination in MEF WT cells (Figure 4a) in comparison to Huh7

cells (Figure 2).

It has recently been shown by super resolution microscopy that lysosomes indeed bind to the PVM and that for normally developing parasites an equilibrium of lysosome binding to the PVM is reached (Niklaus, 2019, Scientific Reports). Association of lysosomes with the PVM is also strongly supported by the fact that the PVM becomes LAMP1 positive in WT infected cells. How do the authors explain this if UIS3-LC3 blocks lysosome binding?

It is very important that they support their model presented in Figure 6 by carefully analyzing lysosome dynamics before and after C4 treatment.

Their ITC experiments indicate that C4 competes for the UIS3-LC3 interaction. These data should be confirmed by more direct binding assays. In their previous publication (Real et al., 2018, Nature Microbiology), they showed a direct binding of UI3 and LC3 by Co-IP experiments. Similar experiments should be done and cells should be treated with C4 to show disruption of this binding.

Reviewer #2 (Remarks to the Author):

The paper uses virtual library screening to select compounds with putative ability to bind at the PfUIS3-hLC3 interacting region.

From 21 compounds identified in the virtual screen, they test 15 compounds for inhibitory activity of Plasmodium sporozoite infection of human hepatocytes. For this, two *P. berghei* lines were used, the WT and a newly generated line that expresses the PfUIS3 in place of the endogenous PbUIS3. These experiments identify C4, a compound able to inhibit spz infection with IC50 at the 90-200 nM range. Further experiments show that C4 binds to the UIS3-LC3 complex, inhibits liver stage development early after sporozoite invasion, and has inhibitory activity only when the host cell has an active autophagy pathway.

The paper shows original results with well conducted experiments, convincingly showing that disrupting the parasite evasion strategy from host cell autophagy inhibits the parasite growth. This is of interest not only to the malaria community as a whole, but also to a more broad audience in fields working with biology and drug discovery against intracellular parasites.

Minor comments:

- I had the impression that the calculations of the IC50s could be better performed had more compound concentrations been tested, producing a complete sigmoid function.
- Figure 1E suggests C13 increases by 8x infection of Huh7 cells with the PfUIS3-expressing Pb sporozoites. In this result consistently reproduced? Is it indeed different to what is seen with the WT Pb spz? Can the authors comment on that?

Reviewer #3 (Remarks to the Author):

The authors have previously shown that Plasmodium UIS3 can bind to host hepatocyte LC3 to protect parasites from the host autophagic response to invasion. They had speculated that this might be a druggable target. In this follow-up manuscript the authors use a virtual compound screening library to identify drugs that bind this UIS3-LC3 target. They focus on the activity of the promising compound (C4) that is shown to target both the falciparum and berghei UIS3 binding to LC3. They characterise the binding using isothermal calorimetry analysis, demonstrate the dose dependence of C4, and importantly show that C4 does not disrupt the normal autophagic response in their HeLa cell model.

This really fascinating manuscript identifies a potential novel therapeutic approach to malaria prophylaxis and possibly an approach that can be used against other intracellular parasites. These results will be of great interest to the malaria community and beyond.

The experimental approaches described are well-described, thorough and systematic. The statistical analyses are appropriate. I have no major suggestions for improvement.

Some minor changes/thoughts:

There appears to be exoerythrocytic forms present even at the highest concentration of C4 tested (10 micromolar; Fig 2a, 2b and 4a). Are these parasites alive? Do you think these parasites are still able to initiate blood stage infections *in vivo*? Perhaps you could comment on this briefly.

Perhaps instead of "ruining" the parasite's ability to evade the host autophagy response, you could use "inhibiting" or "disrupting".

In the text reference is made to mouse parasitaemia and disease progression being shown in supplementary figure 1 panels S1f and S1g respectively, but in my version, these are supplementary figures S1g and S1h respectively.

Point-by-point response to the reviewers

We want to thank the reviewers for the time and effort dedicated to offer valuable feedback on the manuscript. We have now responded to all the insightful comments and incorporated changes in the manuscript to reflect the provided suggestions. All the changes made have been highlighted within the manuscript.

Reviewer #1

Setua and colleagues present an interesting screen for small molecules that interrupt the Plasmodium UIS3-host LC3 interaction. They performed an in-silico screening of more than 20 million compounds and identified 21 compounds that have the potential to bind the UIS3-LC3 interacting region. 15 compounds were tested in a phenotypic screen and one, termed C4, was found to block parasite development in vitro. Further in silico analysis confirmed C4 binding to a pocket that partially overlaps the UIS3-LC3 interacting region. C4 was found to significantly reduce parasite numbers in vitro. When host cell autophagy was blocked by genetic deletion of Atg5 or by chloroquine, the effect was rescued suggesting that C4-mediated elimination of parasites depends on host cell autophagy. The results of this study are promising but preliminary in many respects.

We thank the reviewer for the helpful suggestions and criticisms. We appreciate her/his effort to improve our manuscript, and believe we have now done so.

Major concerns

Comment 1:

In silico analysis revealed 21 compounds but, in the end, only 1 had a consistent effect on the different parasite strains tested. This shows the strength and the weakness of in silico screening at the same time. The strength is certainly that they were able to identify a potent compound. The weakness is that in most other cases the prediction has been wrong. My question is now how can the authors be sure that C4 really binds as they predict? Did they perform the in-silico docking experiments like presented for C4 in Figure 1f and g also for all other 14 compounds and did these fail to bind? In particular, according to their results presented in figure 1c and d, compound C2 should only bind to PbUIS3 and C3 only to PfUIS3. I find it also very strange that they compare luciferase assays with EEF numbers. They should perform comparable assays for both parasite strains that allow a proper comparison.

Response:

Thanks for the suggestion. We sincerely think that the in silico virtual screening is an excellent strategy for functional targeting of druggable molecules, and it harbours more advantages than weaknesses¹. Our work is a good example of such advantages – it showed a high success rate for lead discovery with a limited wet-lab implementation, as stated by the reviewer.

The strategy for the virtual screening was described in the manuscript, but in essence, involves the docking of an open-access database of chemical compounds into UIS3 protein limiting the searching space to the region already characterized as involved in the interaction with LC3². The list of positive hits was ordered by the values of the corresponding DeltaG function, and the ones with the best values selected for further analysis. All the compounds tested in the biological assay (results depicted in Fig. 1c, d and e) were docked to UIS3 structure either in *P. falciparum* or in *P. berghei*. Fig. 1f and g only depicted the predicted interactions between UIS3 and compound C4 for simplification purposes. We have revised the text to make sure all the information is clearly provided.

Regarding the following question about how we are sure about the binding of each compound, we have to consider that our initial validation readout is based on a cellular model where host cells are infected with the parasite in the presence of the analysed drugs (Fig. 1). The binding of each compound to UIS3 and the consequent disruption of the complex is based on two evidences: first, the prediction of the binding site to UIS3 by virtual docking, and second, the readout of the biological assay. If a compound is predicted to bind UIS3 in the described interface of the UIS3-LC3 complex and produces a positive readout in the cellular model assay, it implies that by disrupting the UIS3-LC3 complex it interfere with the parasite survival by an autophagy-mediated mechanism. Indeed, our data in Fig 4, where the effect of the compound was rescued by blocking the host cells autophagy using genetic deletion of Atg5 and LC3 or chloroquine, proves this statement. Results showed in Fig. 1c, d and e are derived from the biological cell assay of activity and are not only related to the binding capacity of each compound to UIS3. Many compounds failed to produce a positive readout (as observed in C5 for instance), meaning that they were not able to interfere with the UIS3-LC3 interaction under these conditions efficiently, and not necessarily that they are unable to interact with UIS3. Other factors also need to be considered, namely those related to the chemical properties of each compound such as solubility and the partition coefficient that could condition their activities. Also, we cannot disregard the existence of molecular interactions of these compounds with other cellular components that could reduce their availability for the specific disruption of LC3-UIS3 interaction. However, the detailed analysis of these additional chemical phenomena are outside of the main scope of the present work.

As suggested by the reviewer, we repeated the compound screen in *P. berghei* using flow cytometry (Fig. 1c) so that the results are comparable with the compound screen in PfUIS@Pb (Fig 1d and e), which were also performed using flow cytometry. We found that the C2, C3 and C4 decreased parasite infection in both *P. berghei* and PfUIS@Pb and confirmed C4 is the most active compound among these three.

Comment 2:

Their statement: “As predicted, C4 binds in a pocket that partially overlaps the proposed UIS3-LC3 interacting region for both *P. falciparum* and *P. berghei* UIS3 (Fig. 1f and g).” is clearly overinterpreted and it indeed needs UIS3-C4 co-crystallization experiments to confirm their assumption. It is very surprising that C4 does not directly interact with any of the key interacting amino acids identified earlier (Real et al. 2018, Nature Microbiology). Has any of the mutants described in the Real paper been tested for C4 interaction? They should extend their in-silico analysis and define mutations that do not allow C4 binding to either PfUIS3 or PbUIS3. These UIS3 mutations should then be generated and expressed recombinantly and tested for C4 binding.

Response:

We agree with the reviewer comment regarding the overstating of the sentence, that probably misleads the reader. In fact, the sentence referred to the fact that the predictions derived from the virtual docking experiments and further solution refinement are compatible with binding of compound C4 into a protein pocket that partially overlaps the already published putative LC3-UIS3 interaction interface. We have now rephrased the sentence: “Docking experiments predicted a binding pocket for C4 in the surface of UIS3 protein that partially overlaps the proposed UIS3-LC3 interacting region of both *P. falciparum* and *P. berghei* UIS3 (Fig. 1f and 1g).

Regarding the proposed co-crystallization experiments to determine the structure of UIS3 complexed with C4, we sincerely believe that they are out of the scope of this manuscript, being more appropriate for a real structural biology work. Moreover, the inherent empiric nature of the crystallization phenomenon could prevent the formation of co-crystals even in the case of positive interaction between UIS3 and C4.

Regarding the reviewer’s observation about the interaction between C4 and the residues of UIS3, we have prepared two supplementary figures that are included in the revised version of the manuscript (Supplementary Figure 4 and 5). In these figures, we present a complete structural analysis of the UIS3-LC3 complex surface interface previously published², and the individual predicted interactions established between C4 and both PbUIS3 and PfUIS3 as determined by virtual docking and refinement. Due to the limitations of space and the general scope of the manuscript, our previous work published in Nature Microbiology² contained a minimal description of the UIS3-LC3 interfaces where the primary interaction pockets were defined by considering the pocket-centred amino acids establishing strong chemical bonds between LC3 and UIS3 (hydrogen bonds) and the nearby residues in each polypeptide chain. In that paper, we did not describe the remaining structure of the interaction pocket, which is comprised of many other amino acids that establish weaker interactions between UIS3 and LC3, mainly based on hydrophobic interactions. We also understand that the small “f” and “g” panels in Figure 1 are not illustrative enough to show that the compound C4 is interacting with some of the previously described residues in UIS3-LC3 complexes. In detail, the revised version of the manuscript contains a Supplementary Figure 4 depicting a plain two-dimensional diagram that includes all the interactions involved in the predicted UIS3-LC3 complex in *P. berghei* and *P. falciparum* and an additional Supplementary Figure 5 representing a two-dimensional plot of compound C4 and the predicted interactions with PbUIS3 and PfUIS3.

By analyzing the data depicted in the new supplementary figures, we can state that in the case of the binding of C4 to PfUIS3 (Suppl. Figure 5, panel a), it interacts with Asn181 by a hydrogen bond established with the nitrile group in C4 and also with Gln217 by hydrophobic interaction. Both amino acids were already described as key residues in UIS3-LC3 interaction in *P. falciparum* in Real *et al.* manuscript. Another vital interaction involves Tyr220 that establishes a predicted pi-pi interaction with an aromatic ring in the main skeleton of C4. Tyr220 was not described as an interacting residue in the previously published work, due to the space limitations of the publication, but it is also an amino acid present in the UIS3-LC3

complex interface establishing hydrophobic interactions with LC3 as depicted in Suppl. Figure 4. Predicted C4-PfUIS3 complex involves an additional interaction comprising a residue not related with the interface of UIS3-LC3 putative complex, which is the cation-pi interaction between Lys218 and compound C4.

Regarding the residues involved in the predicted interaction between C4 and PbUIS3 (See Suppl. Figure 5, panel b), Lys209 was also described in our previous publication as involved in LC3-UIS3 interaction. This residue is predicted to interact with compound C4 by a cation-pi contact (Suppl. Figure 4, Panel b). Asn172 is also predicted to be involved in a hydrogen bond between UIS3 and LC3 proteins in *P. berghei* complex. The predicted interaction map between compound C4 and PbUIS3 is completed by a pi-pi interaction between Tyr212 and the central aromatic ring in C4 (See Suppl. Figure 5, panel b).

We did not check the mutations described in previous work by Real and co-workers for an in silico virtual docking analysis since the objectives of the work described in the present manuscript were focused on the disruption of the functional interactions of UIS3 and LC3 proteins as an antimalarial target. Our work now used the already available knowledge about the key residues putatively involved in UIS3-LC3 interaction to disrupt it, employing a vertical strategy to narrow and reduce the experimental lab work by in silico filtering of chemical candidates.

Comment 3:

They clearly need to show that C4 treatment indeed blocks LC3 labeling of the PVM. I suggest to treat GFP-LC3 expressing *P. berghei*-infected Huh7 cells with C4 and analyze loss of LC3 binding to the PVM.

Response:

Treatment with C4 does not lead to loss of LC3 binding to the PVM. And in fact, based on what we have previously published, this is not unexpected. Indeed, we have previously shown that the PVM of cells infected with parasites fully deficient on UIS3 still show a PVM with the same level of LC3². This led us to conclude that UIS3 does not recruit host LC3 to the PVM. Thus, the fact that C4 treatment does not lead to a reduced level of LC3 at the PVM is fully expected.

Comment 4:

They also should do IFAs to analyze the PVM after C4 treatment using common PVM marker proteins like UIS4 or Exp1. This should give a clear idea whether the PVM is compromised.

Response:

The IFA data was already presented in the submitted version of the paper, Fig. 2h. The red in these images represent PVM protein UIS4. The staining patterns of UIS4 in control and C4 treated cells are very similar, which confirms that C4 treatment does not seem to compromise at least the recruitment of certain PVM-resident proteins.

Comment 5:

Another very important experiment is to infect LC3-deficient cells and treat them with C4, which should not have an additional effect on parasite survival. It would also be very important to test whether UIS3 deficient parasites can normally develop in LC3-deficient cells.

Response:

We have previously shown that UIS3-deficient parasites, while unable to survive in wild-type cells, are perfectly fit in cells depleted of the LC3 conjugation machinery or of Rab7. Cells depleted of ATG3, ATG5, or ATG7, core components of the LC3 conjugation system, are unable to attach LC3 to target membranes^{3,4,5}. The rationale for monitoring infection in cells depleted of the LC3 conjugation machinery, instead of LC3 itself, is precisely to be able to answer whether the lipidation of LC3 was required for the targeting of the molecule to the PVM. Confirming that this is the case, we have also previously shown that parasites infecting *Atg3* and *Atg5* knockout cells show only residual levels of PVM-associated LC3², which confirmed a previous report by Prado *et al.*, in the case of *ATG5*⁶. Thus, to test whether UIS3 deficient parasites can normally develop in LC3-deficient cells, where autophagy pathway is impaired is not necessary and out of the scope of this paper.

We followed the reviewer's suggestion to check if C4 still show activity in LC3-deficient cells. The results clearly show that unlike what happens in WT cells, C4 cannot exert anti-plasmodial effect in LC3 deficient cells. This data is now included in the revised version of the manuscript and shown in Fig. 4b.

Comment 6:

A crucial experiment they describe is the infection and treatment of autophagy-deficient *Atg5*^{-/-} cells as this provides genetic indication that autophagy is eliminating intracellular parasites. However, MEF cells are very poor host cells for *P. berghei* as *P. berghei* does not develop normally in this type of cells. Given the importance of this experiment, they should use better suited *Atg5*-deficient host cells. They also need to show that in C4-treated WT cells, parasites end up in an autophagosome and are indeed eliminated by autophagy. In support of this notion, C4 has a rather modest effect on parasite elimination in MEF WT cells (Figure 4a) in comparison to Huh7 cells (Figure 2).

Response:

As suggested by the reviewer, we now show that C4 activity is also lost in HeLa cells deficient for *Atg5* (Fig. 4a).

Both experiments showing that C4 is unable to eliminate parasites not only in autophagy-deficient HeLa cells (presented above) as well as in cells treated with with the lysosomal inhibitor chloroquine (Fig. 4c and 4d), leads us to conclude that autophagy is involved in the elimination of wild-type parasites in wild-type cells treated with C4. On rare occasions - we have previously published one example² - we were able to observe *uis3*-deficient parasites that appeared to have undergone fusion with Lamp1-positive lysosomes and showing signs of compromised PVM integrity. However, these are very infrequent, suggesting that once fusion occurs, the parasites disappear very rapidly. Still, as suggested by the reviewer (next comment) we have now performed experiments showing that the dynamics of LAMP1

accumulation around the PVM in the presence of C4 is very different from that in non-treated infected cell (see below).

Comment 7:

It has recently been shown by super resolution microscopy that lysosomes indeed bind to the PVM and that for normally developing parasites an equilibrium of lysosome binding to the PVM is reached (Niklaus, 2019, Scientific Reports). Association of lysosomes with the PVM is also strongly supported by the fact that the PVM becomes LAMP1 positive in WT infected cells. How do the authors explain this if UIS3-LC3 blocks lysosome binding? It is very important that they support their model presented in Figure 6 by carefully analyzing lysosome dynamics before and after C4 treatment.

Response:

We thank the reviewer for this suggestion. Indeed, we have now performed experiments to show LAMP1 accumulation around the PVM in the presence or not of C4. The results clearly show that its dynamics is distinct in the presence or absence of C4. The observed association of lysosomes with the PVM is indeed a fascinating – possibly necessary⁷ but conceivably also deleterious in certain conditions or above a certain level – aspect of Host-Plasmodium interactions during the liver stage of infection. These data have now been fully included (Fig. 5a and 5b) and discussed in the revised version of the manuscript.

Comment 8:

Their ITC experiments indicate that C4 competes for the UIS3-LC3 interaction. These data should be confirmed by more direct binding assays. In their previous publication (Real et al., 2018, Nature Microbiology), they showed a direct binding of UIS3 and LC3 by Co-IP experiments. Similar experiments should be done and cells should be treated with C4 to show disruption of this binding.

Response:

These are indeed very interesting experiments but technically very challenging. To visualize UIS3-LC3 interaction, cells are transfected with UIS3, which leads to a high expression of this molecule and a direct visualization of the UIS3-LC3 interaction by IP. Such a high and variable concentration of UIS3 requires a much higher dose (and again variable from experiment to experiment) of C4 that at certain levels becomes toxic to the host cells. Thus, we would prefer not to include these experiments as they are very difficult to reproduce.

Reviewer #2

The paper uses virtual library screening to select compounds with putative ability to bind at the PfUIS3-hLC3 interacting region.

From 21 compounds identified in the virtual screen, they test 15 compounds for inhibitory activity of Plasmodium sporozoite infection of human hepatocytes. For this, two *P. berghei* lines were used, the WT and a newly generated line that expresses the PfUIS3 in place of the endogenous PbUIS3.

These experiments identify C4, a compound able to inhibit spz infection with IC50 at the 90-200 nM range. Further experiments show that C4 binds to the UIS3-LC3 complex, inhibits liver stage development early after sporozoite invasion, and has inhibitory activity only when the host cell has an active autophagy pathway.

The paper shows original results with well conducted experiments, convincingly showing that disrupting the parasite evasion strategy from host cell autophagy inhibits the parasite growth. This is of interest not only to the malaria community as a whole, but also to a more broad audience in fields working with biology and drug discovery against intracellular parasites.

We wish to thank this reviewer for the positive feedback. We have now fully addressed the points raised, which have certainly contributed to improve the clarity of the message.

Minor comments:

Comment 1:

- I had the impression that the calculations of the IC50s could be better performed had more compound concentrations been tested, producing a complete sigmoid function.

Response:

We have performed the experiments again for both *P. berghei* and PfUIS3@Pb parasite lines using flow cytometry. As suggested by the reviewer, we have included more compound concentrations and calculated the IC₅₀. In the revised version of the manuscript, we have presented the data as a sigmoidal curve (Fig. 2a and 2b).

Comment 2:

- Figure 1E suggests C13 increases by 8x infection of Huh7 cells with the PfUIS3-expressing Pb sporozoites. In this result consistently reproduced? Is it indeed different to what is seen with the WT Pb spz? Can the authors comment on that?

Response:

In the supplementary material of the revised version of the manuscript, we now present the results of three individual experiments with C13 in PfUIS3@Pb (Supplementary Figure S6). The results confirm that in PfUIS3@Pb line, this compound always increased the infection compared to the control, which as pointed by the reviewer is not the case in *P. berghei* sporozoites. This is indeed an intriguing result but probably not unexpected. Our *in silico* analysis show that the interacting amino acids in *P. berghei* UIS3 and *P. falciparum* UIS3 are not exactly the same opening the possibility that a particular compound, in this case C13, may interfere with the interaction of each of these molecules with LC3 in different ways.

Reviewer #3

The authors have previously shown that Plasmodium UIS3 can bind to host hepatocyte LC3 to protect parasites from the host autophagic response to invasion. They had speculated that this might be a druggable target. In this follow-up manuscript the authors use a virtual compound screening library to identify drugs that bind this UIS3-LC3 target. They focus on the activity of the promising compound (C4) that is shown to target both the falciparum and berghei UIS3 binding to LC3. They characterise the binding using isothermal calorimetry analysis, demonstrate the dose dependence of C4, and importantly show that C4 does not disrupt the normal autophagic response in their HeLa cell model.

This really fascinating manuscript identifies a potential novel therapeutic approach to malaria prophylaxis and possibly an approach that can be used against other intracellular parasites. These results will be of great interest to the malaria community and beyond. The experimental approaches described are well-described, thorough and systematic. The statistical analyses are appropriate. I have no major suggestions for improvement.

We want to thank the referee for the positive insightful comments, which we believe have helped us improve the quality of the manuscript.

Some minor changes/thoughts:

Comment 1:

There appears to be exoerythrocytic forms present even at the highest concentration of C4 tested (10 micromolar; Fig 2a, 2b and 4a). Are these parasites alive? Do you think these parasites are still able to initiate blood stage infections in vivo? Perhaps you could comment on this briefly.

Response:

This is a very interesting point. The evidence suggests that some parasites may survive C4 treatment. In fact, it has been previously shown both in mice and in humans that infection with uis3-deficient parasites show frequent breakthroughs⁸. Not all wild type cells present the same autophagy capacities and, as our results clearly show, C4 is only active in autophagy-competent cells. These thoughts and concept have now been included in the discussion of the revised version of the manuscript.

Comment 2:

Perhaps instead of “ruining” the parasite’s ability to evade the host autophagy response, you could use “inhibiting” or “disrupting”.

Response:

In the revised version of the manuscript, we have altered this word as the reviewer suggested.

Comment 3:

In the text reference is made to mouse parasitaemia and disease progression being shown in

supplementary figure 1 panels S1f and S1g respectively, but in my version, these are supplementary figures S1g and S1h respectively.

Response:

We apologise for our mistake, which has been now corrected in the revised version of the manuscript.

1. Gorgulla, C. *et al.* An open-source drug discovery platform enables ultra-large virtual screens. *Nature* **580**, 663-668 (2020).
2. Real, E. *et al.* Plasmodium UIS3 sequesters host LC3 to avoid elimination by autophagy in hepatocytes. *Nature microbiology* **3**, 17-25 (2018).
3. Kuma, A. *et al.* The role of autophagy during the early neonatal starvation period. *Nature* **432**, 1032-1036 (2004).
4. Sou, Y.S. *et al.* The Atg8 conjugation system is indispensable for proper development of autophagic isolation membranes in mice. *Molecular biology of the cell* **19**, 4762-4775 (2008).
5. Lee, I.H. *et al.* Atg7 modulates p53 activity to regulate cell cycle and survival during metabolic stress. *Science* **336**, 225-228 (2012).
6. Prado, M. *et al.* Long-term live imaging reveals cytosolic immune responses of host hepatocytes against Plasmodium infection and parasite escape mechanisms. *Autophagy* **11**, 1561-1579 (2015).
7. Lopes da Silva, M. *et al.* The host endocytic pathway is essential for Plasmodium berghei late liver stage development. *Traffic* **13**, 1351-1363 (2012).
8. Kumar, H. *et al.* Protective efficacy and safety of liver stage attenuated malaria parasites. *Scientific reports* **6**, 26824 (2016).

REVIEWERS' COMMENTS:

Reviewer #1 (Remarks to the Author):

The authors have carefully addressed all my concerns and I can now recommend the manuscript for publication.

Reviewer #2 (Remarks to the Author):

All my concerns have been addressed.
I think this is an excellent manuscript.

Reviewer #3 (Remarks to the Author):

This revised version of the original manuscript displays additional data and explanations in the text that, in my opinion, have strengthened the authors' hypotheses and evidence.
I am satisfied with the authors' responses to my queries regarding the original submission.